# OpenReview forum: "Chinese Tiny LLM: Pretraining a Chinese-Centered Large Language Model"
_colmweb.org/COLM/2024/Conference — COLM_

### Official Review · Reviewer_fKZj · 2024-05-09

**Rating:** 5
**Confidence:** 5
**Ethics Flag:** 1

**Summary:**

The paper introduces the Chinese Tiny LLM (CT-LLM), a large language model specifically designed for Chinese language processing, diverging from traditional models by primarily incorporating Chinese textual data. This model is pretrained on a mixed corpus of 1.2 trillion tokens, predominantly Chinese, but also including English and code tokens, aiming to excel in both Chinese language tasks and multilingual capabilities. The study emphasizes the shift away from models predominantly trained on English, showcasing CT-LLM’s performance across several benchmarks, particularly highlighting its proficiency in Chinese with tools like the Chinese Hard Case Benchmark (CHC-Bench). By open-sourcing the entire process and methodologies used for training CT-LLM, the paper aims to promote further research and diversity in the development of language models across different linguistic and cultural contexts.

**Questions To Authors:**

1. At the end of Section 1, the authors claim that they propose "the first Chiense-centered large language model", while there are also Baichuan, Qwen, ChatGLM, etc. focusing on Chinese. The authors should justify this, as section 2.1 does not fully support this claim.
2. At the end of Section 4, the authors said that "In our data filtering process for the SFT dataset, we retain only those entries with a perplexity score below 3,000 under Qwen-7B." Actually, 3000 PPL is extremely high, is this a reasonable threshold?



Other comments:

1. Section 2.1: many citation formats are wrongly used. For example, Mistral Jiang et al. (2023) => Mistral (Jiang et al., 2023).
2. Section 6 results: CHC-Benchwith => CHC-Bench with

**Reasons To Accept:**

1. The paper proposes a small LLM, called CT-LLM, which might be useful in resource-scarce scenarios.
2. The pretraining data processing pipeline as well as its open-source resources may facilitate other researchers working on Chinese LLM construction.

**Reasons To Reject:**

1. The model design is a combination of previous techniques, leading to less novelty. Also, the pre-training/SFT/DPO scheme is quite normal.
2. The experimental results are not good enough. For example, in Table 2, the proposed CT-LLM significantly lags behind other baseline models. While it can achieve better performance than other English-majored LLMs on C-Eval and CMMLU, the improvements are quite moderate and significantly legs behind Qwen. Similar observation also happens in Table 3 and 5.
3. The proposed CHC-Bench is rather small. According to Appendix C.3., it only has 214 samples with ~30 samples for each category. However, in this context, the proposed CT-LLM still can not outperform other baseline models, with many results ranked 3rd or below, where these results are significantly far from 1st/2nd ranked models. Combined with weakness #2, I am not convinced that the proposed model performs well.

---

> ### Author Rebuttal · Authors · 2024-05-31
>
> Thanks so much for your review.
>
> # Reject 1
> Please refer to the response to Reviewer Rmg1 Reject 2
>
> # Reject 2
> Please refer to the response to Reviewer Rmg1 Question 1
>
> # Reject 3
> Our CHC-BENCH is designed similarly to MT-Bench and faces constraints due to the manual curation of high-quality questions and human or GPT4 evaluations. We optimized data volume for cost-effectiveness, taking balance with MT-Bench's 80 samples across 8 domains and Alpaca-Eval's 802 samples from existing benchmarks.
>
> For performance-wise, our model outperforms many previous open-source models. The remaining gaps may be due to the lack of some high-quality data that is not publicly available. Instead, we have ensured reproducibility by releasing all training data and methods. Under the right circumstances, our research on the Chinese model pre-training can be generalized to higher-quality training corpora.
>
> # Question 1
> Our assertion that "our model is Chinese-centered" is substantiated by the detailed breakdown of our pretraining data distribution illustrated in Figure 1. Specifically, Chinese data constitutes 66.99% of the total. In currently available pre-trained models, the proportion of Chinese data does not exceed 50%, including the Yi series, which specializes in Chinese. Moreover, none of the currently known models claim to be a Chinese-centered large model.
>
> # Question 2
> We identified a typo in our code that led to unusually high perplexity (PPL) scores. However, this did not affect the final results as the entries were correctly ordered based on their relative PPL values.
>
> We fixed the error, and report the new ppl at the following [anonymous link](https://www.upload.ee/files/16687882/ppl_coig_cqia.json.html)
>
> Mean PPL: 19.15.
> Median PPL: 12.38.
> 25th Percentile (Q1) PPL: 8.24.
> 75th Percentile (Q3) PPL: 20.40

---

> ### Comment · Reviewer_fKZj · 2024-06-07
>
> Thanks for the rebuttal. I decide to keep my original rating, due to 1) I am not convinced by the novelty of the proposed model; 2) rather moderate experimental results.

---

### Official Review · Reviewer_n9Pq · 2024-05-11

**Rating:** 6
**Confidence:** 5
**Ethics Flag:** 1

**Summary:**

This paper presents CT-LLM, a 2B parameter language model that takes an approach of centering Chinese language in its pretraining. The model is trained from scratch on a corpus of 1,200B tokens, with a composition of 800B Chinese tokens, 300B English tokens, and 100B code tokens. Supervised fine-tuning and preference optimization are applied to further enhance the model's capabilities. The paper introduces valuable resources including the MAP-CC pretraining corpus, CHC-Bench evaluation benchmark, and the open-sourced CT-LLM model. Results demonstrate CT-LLM's strong performance on Chinese tasks, competitive English abilities after supervised fine-tuning, and distinct characteristics compared to English-centric models.

Here is my review of the new paper "Chinese Tiny LLM: Pretraining a Chinese-Centered Large Language Model":
<review>
Summary: This paper presents CT-LLM, a 2B parameter language model that takes a novel approach of centering Chinese language in its pretraining. The model is trained from scratch on a corpus of 1,200B tokens, with a composition of 800B Chinese tokens, 300B English tokens, and 100B code tokens. Supervised fine-tuning and preference optimization are applied to further enhance the model's capabilities. The paper introduces valuable resources including the MAP-CC pretraining corpus, CHC-Bench evaluation benchmark, and the open-sourced CT-LLM model. Results demonstrate CT-LLM's strong performance on Chinese tasks, competitive English abilities after supervised fine-tuning, and distinct characteristics compared to English-centric models.
The paper makes several meaningful contributions:
- It challenges the prevailing paradigm of training LLMs primarily on English data and then adapting to other languages. Instead, it explores a Chinese-centric approach from the ground up.
- The MAP-CC corpus and detailed data processing pipeline provide high-quality Chinese pretraining data and an effective methodology for the community.
- The CHC-Bench offers a comprehensive and multidisciplinary benchmark for evaluating Chinese language understanding and reasoning capabilities.

Open-sourcing the full training process, data, code, and model promotes reproducibility and further innovation.
The experiments are extensive, covering automated evaluations on Chinese and English benchmarks, human evaluations via a chatbot arena, analysis of political biases, and probing of model components. The results convincingly demonstrate CT-LLM's strong Chinese language abilities, competitive English performance post-finetuning, and intriguing differences from prevailing models.

**Reasons To Accept:**

- Novel Chinese-centric pretraining approach that challenges English-dominant paradigms
- Releases valuable resources in MAP-CC corpus, CHC-Bench, and open-sourced model
- Strong empirical results on CT-LLM's Chinese and English capabilities
- Insightful analyses on political biases and model components
- Significant implications for cross-lingual LLM development and language inclusivity

**Reasons To Reject:**

- Model size (2B parameters) is limited compared to current state-of-the-art LLMs (30B+ parameters), raising questions about the scalability and ultimate ceiling of the Chinese-centric approach.
- Analysis of CT-LLM's limitations and failure modes is absent. Understanding where and why the model struggles is important for contexualizing its capabilities.
- Efficiency metrics like training time, inference latency, and memory footprint are not reported, leaving open questions about practical applicability.

---

> ### Author Rebuttal · Authors · 2024-05-31
>
> Thanks so much for your review.
>
> # Limited Model Size
> Please refer to the response to Reviewer Rmg1 Reject 1
> # Model Limitation and Failure Modes
> Unlike most prior efforts that adopt an English-centric pretraining paradigm, CT-LLM explores the potential of open-source non-English-centric pretraining.
>
> First, the model shows improved cross-lingual generalization, particularly on English benchmarks like WinoGrande and Hellaswag, and performs impressively on English math, code, and knowledge benchmarks. The mechanisms behind cross-lingual generalization, whether English-centric or non-English-centric, require further investigation.
>
> Second, we will explore more Chinese corpora to enhance non-English-centric pretraining. For reference, DeepSeek-v2 (not released when CT-LLM was submitted) uses over 3 trillion Chinese tokens, twice as many as MAP-CC.
>
> Third, with additional financial support, we aim to scale up non-English-centric pretraining.
>
> # Efficiency metrics like training time, inference latency, and memory footprint are not reported, leaving open questions about practical applicability.
> Our model was trained using 384 H800 GPUs over the course of 2 days, with training lasting for 190K steps. The tokens per second per GPU reached approximately 20,000.
>
> For the inference phase, testing was conducted on a single H800 accelerator, utilizing an input sequence of maximum length 4096, under bfloat16 precision; the latency observed for the first token inference was 0.832 seconds with a Memory Footprint Utilization (MFU) of 20%.
>
> For "tokens per second per GPU over steps" and "Train loss over steps", please refer to [link](https://www.upload.ee/files/16693619/metric.zip.html)

---

> ### Author Response · Authors · 2024-06-07
>
> Dear Reviewer n9Pq,
>
> Could you please review our response to see if we have addressed your comment effectively? Specifically, we have clarified our explanations regarding model size, limitations, failure modes, and efficiency metrics such as training time, inference latency, and memory footprint.
>
> If you find that our revisions effectively address your concerns, we kindly ask if you could consider adjusting your comments or score accordingly.
>
> Thank you very much for your time and consideration.
>
> Best regards,
>
> Authors

---

### Official Review · Reviewer_bTrV · 2024-05-16

**Rating:** 7
**Confidence:** 3
**Ethics Flag:** 1

**Summary:**

The paper introduces CT-LLM, a large-scale language model explicitly designed for the Chinese language.
This model diverges from previous efforts that predominantly utilized English data sets by conducting training from scratch, emphasizing Chinese while also incorporating English and code tokens. Techniques such as SFT are utilized to boost performance in both Chinese and English.

**Questions To Authors:**

* The most intriguing part of the paper is its evaluation of political biases. This aspect is both new and interesting. However, the paper fails to explain how political biases, such as authoritarian and libertarian tendencies, are assessed. Is the evaluation based solely on the authors' observations? It would be highly interesting if the authors could devise a method for evaluating political biases.

* Figure 3 is not very clear to me.

* In Table 2, the definition of HumanEval (pass@1) is unclear. Additionally, why does Phi-2 have a score of 0 for HumanEval? This is peculiar, given that Phi-2's other metrics are quite strong.

* The distinction between the experiments in Table 2 and Table 3 is unclear. It appears that the experiment in Table 3 also incorporates English training data?

**Reasons To Accept:**

The primary contributions of CT-LLM include

1. a high-quality Chinese corpus
2. a benchmark that evaluates Chinese hard cases
3. the model

The paper details the training procedure, making it a valuable manual for researchers aiming to train an LLM from scratch despite not proposing any new technology.

**Reasons To Reject:**

* There is no new technology.
* The most intriguing part of the paper is its evaluation of political biases. This aspect is both new and interesting. However, the paper fails to explain how political biases, such as authoritarian and libertarian tendencies, are assessed. Is the evaluation based solely on the authors' observations? It would be highly interesting if the authors could devise a method for evaluating political biases.

---

> ### Author Rebuttal · Authors · 2024-05-31
>
> Thanks so much for your review.
> # Reject 1
> Please refer to the response to Reviewer Rmg1 Reject 2
> # Evaluation of political biases and Figure 3
> We use a questionnaire consistent with the evaluating human to assess the model's political stance, the questionnaire can be found at [link](https://www.politicalcompass.org/test), the only difference is that the LLM is not able to select the questionnaire options on the web page, and therefore needs to first perform inference by prompting all the questions and then constructing an auto-selection script to execute it.
>
> Figure 3 presents the final results of the evaluation, which contains two main types of evaluation dimensions for political leanings, the horizontal axis labels the left/right leaning of the large model, which implies how radical/conservative the model is, and the closer it converges to the origin implies the more neutral it is. The vertical axis labels the liberal leanings of LLMs, which lower scores imply that the model is less likely to be conformist (more freedom in viewpoint). We would like to point out that there is no superiority or inferiority in political tendencies in this test, but simply reflects the average response strategy that a model may produce when faced with the same problem.
>
> # HumanEval (pass@1) is Unclear
> The pass@k metric measures the likelihood that at least one out of the k code samples generated for each task in the benchmark successfully passes its unit tests. In our evaluation, We set k to 1. This indicates that only one sample for each LLM, per task, will be generated for evaluation on the target tasks.
>
> # Phi-2 has a score of 0 for HumanEval
> we re-ran the experiments using Opencompass on the HumanEval benchmark, specifically for the Phi-2 model. Our re-evaluation revealed that Phi-2's performance was actually 40.24, which is in line with its overall capabilities and other benchmark scores. The initial score of 0 appears to have been a reporting error. The evaluation report can be seen at [link](https://www.upload.ee/files/16693544/phi2_humaneval.txt.html)
>
> # The distinction between the experiments in Table 2 and Table 3
>
> Table 2 represents the comparison of base models, whereas Table 3 represents the chat model that has been further trained through SFT and RLHF.

---

> > ### Comment · Reviewer_bTrV · 2024-06-06
> >
> > Thank you for your response. I would like to increase the score accordingly. Please include the details about the evaluation of political biases and fix the mistakes in Table 2 in your next version.

---

### Official Review · Reviewer_Rmg1 · 2024-05-16

**Rating:** 6
**Confidence:** 3
**Ethics Flag:** 1

**Summary:**

The paper introduces CT-LLM, a 2 billion parameter Chinese-centered large language model pretrained primarily on 800 billion Chinese tokens along with some English and code tokens. The authors claim this represents a shift from the prevailing paradigm of training LLMs mainly on English data. They describe techniques like supervised fine-tuning and preference optimization to enhance CT-LLM's Chinese and English capabilities. The paper also introduces the Massive Appropriate Pretraining Chinese Corpus (MAP-CC), the Chinese Hard Case Benchmark (CHC-Bench) for evaluation, and analyzes CT-LLM's political biases.

**Questions To Authors:**

- The performances of Table 2/3 do not seem strong enough. Where is the gap from?
- How well does CT-LLM generalize to low-resource languages beyond English and Chinese?

**Reasons To Accept:**

- The paper proposes a new idea of pretraining an LLM primarily on Chinese data.
- The paper uses extensive data filtering and deduplication methods to curate a high-quality Chinese pretraining corpus (MAP-CC).
- The paper develops CHC-Bench to evaluate complex skills like instruction following in Chinese.

**Reasons To Reject:**

- The paper trains the LLM with only 2 billion parameters, which constrains its capabilities.
- The model architecture and the tokenizer are the same as those of previous LLMs.

---

> ### Author Rebuttal · Authors · 2024-05-31
>
> Thanks so much for your review.
> # Reject1
> We emphasize the following key points for CT-LLM:1. **Extrapolation to Larger Models**: our line deduplication methods and MAP-CC dataset are used in the 7B MAP-Neo [1], which performs on par with LLama2.
> 2. **Financial Constraints**: Training larger models involves significant costs, to train a 7B model on over 1 trillion tokens can cost nearly one hundred thousand dollars.
> In comparing CT-LLM with larger models, we note the following advantages: computational efficiency, faster inference, and lower financial costs
> # Reject2
> Current LLMs [2][3][4] often use similar tokenizer and model architecture i.e. RoPE for position embedding, RMSNorm for normalization, etc,  with minimal ablation studies on tokenizer effects.
> ## Key Contributions:
> 1. We provide the largest velidated (800B) pretrain dataset on our target language, ensuring high data quality and relevance.
> 2. We open-source an effective processing pipeline for the pretrain data: similar line deduplication and heuristic rules for data filtering.
> 3. We demonstrate pretrain on a non-English dominated corpus can achieve comparable performance and cross-lingual generalization. This encourages the development of non-English corpora, promoting greater inclusivity and attention to diverse linguistic resources.
>
> # Question1
> 1. CT-LLM performs competitively to industrial SOTA LLMs, e.g. Gemma. On target language benchmarks (CMMLU and C-Eval), CT-LLM achieves around +5% better performance than Gemma-2b and Phi-2. On the MMLU English benchmark, CT-LLM performs +12% better than TinyLLama, with only a -4% performance gap compared to the English-centric Gemma-2b.
>
> The performances gaps are largely due to the complex engineering and private data curation used by industrial models to boost specific downstream tasks. Our goal is to investigate our primary research problem, so we have not focused extensively on domain-specific data curation.
> Similar to another transparent LLM, OLMo [5], which also lags behind industrial models like LLama and Pythia, our focus is on providing valuable scientific insights rather than solely competing on performance metrics.
>
>
> # Question2:
> In our scenario, code can be seen as low-resource data. Despite having only 7.91% of pretraining data dedicated to code, we achieve equal performance with Gemma-2b on humaneval.
>
>
> # Reference (Arxiv ID)
> [1] 2405.19327
>
> [2] 2309.16609
>
> [3] 2309.10305
>
> [4] 2307.09288
>
> [5] 2402.00838

---

> > ### Comment · Reviewer_Rmg1 · 2024-06-05
> >
> > Thank you for the response. I would like to keep my score.

---

### Decision · Program_Chairs · 2024-07-10

**Decision:**

Accept

**Comment:**

The reviewers agreed that this paper provides several valuable contributions:
- A high-quality and extensive Chinese corpus (800B).
- A data processing pipeline for filtering pretraining data.
- A benchmark for evaluating Chinese hard cases (CHC-Bench).
- New insights into political bias.
However, some concerns were raised regarding the limited novelty of the model architecture and the relatively small size of the model. Despite these concerns, the paper's contributions—particularly the datasets, tools, and findings—are significant enough to be ready for publication.